# Viral DNA Accumulation Regulates Replication Efficiency of *Chlorovirus* OSy-NE5 in Two Closely Related *Chlorella variabilis* Strains

**DOI:** 10.3390/v15061341

**Published:** 2023-06-09

**Authors:** Ahmed Esmael, Irina V. Agarkova, David D. Dunigan, You Zhou, James L. Van Etten

**Affiliations:** 1Nebraska Center for Virology, University of Nebraska, Lincoln, NE 68583, USA; ahmedesmael@unl.edu (A.E.); irina@unl.edu (I.V.A.); ddunigan2@unl.edu (D.D.D.); 2Botany and Microbiology Department, Faculty of Science, Benha University, Benha 13518, Egypt; 3Department of Plant Pathology, University of Nebraska, Lincoln, NE 68583, USA; 4Center for Biotechnology, University of Nebraska-Lincoln, Lincoln, NE 68588, USA; yzhou2@unl.edu

**Keywords:** chloroviruses, algal viruses, OSy-NE5, *Chlorella variabilis*, restriction, intracellular resistance, DNA accumulation

## Abstract

Many chloroviruses replicate in *Chlorella variabilis* algal strains that are ex-endosymbionts isolated from the protozoan *Paramecium bursaria,* including the NC64A and Syngen 2-3 strains. We noticed that indigenous water samples produced a higher number of plaque-forming viruses on *C. variabilis* Syngen 2-3 lawns than on *C. variabilis* NC64A lawns. These observed differences led to the discovery of viruses that replicate exclusively in Syngen 2-3 cells, named Only Syngen (OSy) viruses. Here, we demonstrate that OSy viruses initiate infection in the restricted host NC64A by synthesizing some early virus gene products and that approximately 20% of the cells produce a small number of empty virus capsids. However, the infected cells did not produce infectious viruses because the cells were unable to replicate the viral genome. This is interesting because all previous attempts to isolate host cells resistant to chlorovirus infection were due to changes in the host receptor for the virus.

## 1. Introduction

Viruses vary from extreme specialists to broad generalists [1]. Some viruses can infect hosts from genetically distinct strains of the same species, whereas others can only infect a single host strain. Still other viruses infect hosts from distantly related species. Intraspecific infectivity of algal viruses is common and has been reported in several instances. For example, (i) ssRNA viruses that infect the dinoflagellate *Heterocapsa circularisquama* [2], (ii) the raphidophyte *Heterosigma akashiwo* and its virus HaV [3], (iii) the viruses of the diatom *Rhizosolenia setigera* [4], and (iv) the dsDNA virus that infects *Phaeocystis pouchetii* [5]. On the other hand, interspecific infectivity is rare among algal viruses; one example is viruses from the *Phaeovirus* genus, which infect the brown alga *Ectocarpus siliculosus* but are also able to infect other algae such as *Kukuckia kylinii* [6,7].

Host resistance to viral infection is crucial for both virus and host population dynamics and survival in natural environments. Several resistance mechanisms have been reported in microbial hosts, including bacteria and phytoplankton. All are associated with a particular stage of viral infection [8], such as attachment, in which the virus physically recognizes and attaches to the host cell; viral entry, during which the virus or its nucleic acid enters the cell; viral gene expression; viral protein translation; viral genome replication; virion assembly; and cell lysis [9]. Attachment of the virus to the host cell is the first step in viral infection, and many hosts escape this step by altering the cell surface receptors or just eliminating them [9]. It has also been reported that some algal species create colonies with an exterior protective membrane or mucus (a biofilm-like structure) that protects them against viral infection [9] via preventing viral attachment or penetration. For example, the virus PpV-AJ96 does not infect the colonial stage of its algal host, *Phaeocystis pouchetii*, which is enclosed in a robust, thin, semi-permeable membrane [10].

The coccolithophore alga *Emiliania huxleyi* has evolved several ways to avoid virus (E. hux viruses or EhV) infection. The alga has two very different phenotypes: the calcified phase, which can lead to massive algal blooms and is diploid, and a non-bloom forming haploid phase [11]. The diploid form is susceptible to EhV viruses and the haploid form, which is physiologically different, is completely resistant to EhV virus infection [12]. *E. huxleyi* can also develop resistance to EhV viruses by altering its external membrane so that the virus cannot attach to it. Finally, programed cell death machinery is also involved in EhV replication and alterations in this process can play a role in virus resistance [13,14,15,16]. Interestingly, intracellular suppression mechanisms have rarely been reported in resistance to algal viruses; however, Tomaru et al. [17] reported restriction of the virus HcRNAV genome replication in the marine dinoflagellate *Heterocapsa circularisquama*. The first intracellular resistance found in *Chloroviruses* is described in this article.

*Chloroviruses* (family *Phycodnaviridae*) are large, plaque-forming, icosahedral viruses that infect certain chlorella-like green algae. They have an internal lipid membrane and linear dsDNA genomes that range in size from 290 to 370 kb, which encode ~330 to 400 proteins (CDSs) and up to 16 tRNAs. Collectively, 35 chloroviruses encode 1345 clusters of orthologous groups of genes (COGs) [18]. Chloroviruses are cosmopolitan residents of inland waters with titers occasionally reaching thousands of plaque-forming units (PFUs) per ml of indigenous waters [19,20,21,22,23,24]. All chlorovirus hosts are normally endosymbionts, often referred to as zoochlorellae [25,26,27]. They are known to be associated with the protozoan *Paramecium bursaria*, the coelenterate *Hydra viridis,* or the heliozoon *Acanthocystis turfacea* [28]. Zoochlorellae are resistant to viruses in their symbiotic state because they are protected by the superhost that provides a physical barrier between the virus and the zoochlorellae. Fortunately, some zoochlorellae grow independently of their partners in the laboratory, permitting plaque assay of the viruses and synchronous infection of their hosts, which allows one to study the virus life cycles in detail [29].

Paramecium bursaria chlorella virus 1 (PBCV-1) is a type member of the genus *Chlorovirus.* PBCV-1 infects and forms plaques on two *C. variabilis* isolates, NC64A and Syngen 2-3 (hereafter referred to as NC64A and Syngen, respectively), both of which are endosymbionts of *P. bursaria* [30]. We assumed that the NC64A and Syngen strains were identical at the time the plaque assay was developed for PBCV-1 in 1983, and consequently our studies focused on the strains of PBCV-1 and NC64A present for the past 40 years [28]. However, a taxonomic study on rDNA [31], as well as physiological analyses [32] of the two *C. variabilis* strains, established that NC64A and Syngen were similar, but not identical organisms. This information prompted us to look for viruses in native water that would form plaque on Syngen lawns. Surprisingly, viruses that form plaques on Syngen are more common in indigenous waters than viruses that form plaques on NC64A at certain times of the year [33]. This observation led to the discovery that some chloroviruses only replicate in Syngen cells (named Only Syngen viruses, OSy viruses) and not in NC64A cells. However, the OSy-NE5 virus can attach to NC64A cells, as determined using a standard attachment assay [33].

Furthermore, OSy-NE5 virus attachment to NC64A cells caused the rapid uptake of SYBR^®^ Gold stain and consequently positive DNA fluorescent staining in the infected cell [33]. This suggested that the virus initiated infection where the host cell membrane was depolarized and that OSy-NE5 DNA entered the cell, however, without completing replication. This result was unexpected because all our previous attempts to isolate virus-resistant chlorella cells were due to changes in the virus receptor such that the chloroviruses could not attach to the host (unpublished results). In this study, we extended these previous investigations to evaluate why OSy-NE5 replication does not progress to completion in NC64A cells.

## 2. Materials and Methods

### 2.1. Viruses and Cell Cultures

*Chlorella variabilis* NC64A and Syngen 2-3 cells were grown on Modified Bold’s Basal Medium (MBBM) with 10 μg/mL tetracycline [30,34]. Cell cultures were shaken (200 rpm) at 25 °C under continuous light, and growth was monitored by direct cell counts with a hemocytometer. PBCV-1 and OSy-NE5 viruses were propagated and purified as described previously [29,30,35].

### 2.2. Exposing NC64A and Syngen Cells to OSy-NE5 and PBCV-1 Viruses

Fresh daily cultures of NC64A and Syngen cells were infected individually with OSy-NE5 at two different multiplicities of infections (MOIs) of 0.001 or 10 plaque-forming units (PFU)/cell. Uninfected cells served as a negative control. Infected and uninfected cell cultures were incubated at 25 °C under continuous light while shaken at 200 rpm. Algal growth was monitored at intervals for 15 days post-infection (PI) by direct cell counts with a hemocytometer. As a positive control, along with this experiment, both cultures of NC64A and Syngen cells were infected with PBCV-1 under the same conditions.

### 2.3. Transmission Electron Microscopy

Cultures of NC64A and Syngen were infected with filter-sterilized viruses (PBCV-1 and OSy-NE5) at an MOI of 5 PFU/cell, and cell cultures were shaken (200 rpm) at 25 °C under continuous light. At appropriate times PI, aliquots of infected cells were sampled and collected via centrifugation in a Sorvall GSA rotor at 5000× *g* for 5 min at 4 °C. The pelleted cells were resuspended and fixed with 2% glutaraldehyde and 2% paraformaldehyde in 0.1 M phosphate buffer (pH 7.4), and further fixed in 1% osmium tetroxide in 0.1 M phosphate buffer (pH 7.4) for 1 h. Samples were dehydrated in a graduated ethanol series and embedded in Epon 812 (Electron Microscopic Sciences, Fort Washington, PA, USA). Ultrathin sections (80 nm) were cut using a Leica UC7 ultramicrotome and stained with uranyl acetate and lead citrate. Ultrastructural images were obtained using a bottom-mount AMT-8 advanced digital camera with a Hitachi H7500 transmission electron microscope at the Morrison Microscopy Core Research Facility in the Center for Biotechnology at the University of Nebraska—Lincoln.

### 2.4. Isolation of DNA and Protein

Actively growing NC64A and Syngen cultures were infected with OSy-NE5 and PBCV-1 at a MOI of 5 PFU/cell. Infected cells were collected at various times PI via centrifugation at 5000× *g* for 5 min, 4 °C, and the pellet fractions were frozen in liquid nitrogen and stored at −80 °C. The harvested cells were homogenized using a Mini-Beadbeater-96 at high speed in TRIzol reagent containing phenol (Ambion^®^ by Life Technologies^TM^, Carlsbad, CA, USA) in the presence of 0.25 mm RNase-free baked glass beads to disrupt the algal cell walls. Chloroform was added to the homogenized mixture, incubated for 3 min at room temperature, and centrifuged at 12,000× *g* for 15 min at 4 °C. The interphase and organic phenol–chloroform phases were collected and used for DNA and protein isolation.

DNA was precipitated from the interphase/organic layer by adding 100% ethanol to reach 70% ethanol and incubated for 3 min at room temperature, followed by centrifugation at 2000× *g* for 5 min at 4 °C. The phenol–ethanol supernatant was collected for protein isolation. DNA pellets were washed with 1 mL of sodium citrate–ethanol solution (0.1 M sodium citrate in 10% ethanol, pH 8.5). After 30 min of incubation at room temperature, the tubes were centrifuged at 2000× *g* for 5 min at 4 °C. A total of 1.5 mL of 75% ethanol was added to DNA pellet fractions and incubated for 20 min at room temperature, and then the tubes were centrifuged at 2000× *g* for 5 min at 4 °C. The supernatant fraction was discarded, and DNA pellet fractions were air-dried for 5 min. DNA pellets were resuspended in 8 mM NaOH at a concentration of 0.2 µg/µL, and the pH was adjusted to 7 with 1 M HEPES. DNA yields were evaluated using a NanoDrop ND-1000 UV-Vis spectrophotometer. DNA samples were stored at −20 °C.

Protein was isolated from the phenol–ethanol supernatant fraction by adding 1.5 mL of 100% isopropanol per 1 mL of TRIzol reagent used in the homogenization and incubated for 10 min at room temperature, and then the sample was centrifuged at 12,000× *g* for 10 min at 4 °C. The supernatant fraction was discarded. Protein pellet fractions were washed 2 times with 0.3 M guanidine hydrochloride in 95% ethanol, incubated for 20 min at room temperature, and then they were centrifuged at 7500× *g* for 5 min at 4 °C. A total of 2 ml of 100% ethanol was added to the pellet fractions, incubated for 20 min at room temperature, and then centrifuged at 7500× *g* for 5 min at 4 °C. The protein pellet fractions were air-dried for 5 min and then resuspended in 1% SDS by incubating at 50 °C in a heat block. After the protein was dissolved, it was centrifuged at 10,000× *g* for 10 min at 4 °C to sediment any insoluble materials, and the supernatant fraction was transferred to a clean tube. Protein was quantified using a Qubit^®^ Protein Assay Kit (ThermoFisher Scientific, Waltham, MA, USA) and read with a Qubit 3.0 Fluorometer as per the manufacturer’s protocol. Protein samples were stored at −20 °C.

### 2.5. SDS-PAGE and Western Blotting

Purified viruses and cell lysate proteins were suspended in 62.5 mM Tris, pH 6.8, 3% (*w*/*v*) SDS, 0.1 M dithiothreitol, 20% (*w*/*v*) glycerol, and 0.02% (*w*/*v*) bromophenol blue and heated at 100 °C for 5 min. Equal protein concentrations (10 mg) were layered onto a linear 4 to 20% Tris-Glycine Mini-PROTEAN^®^ TGX™ Precast Gel (Bio-Rad) and electrophoresed with Laemmli buffer [36]; electrophoresis was performed at a constant voltage of 200 V for about 40 min using a tank buffer consisting of 0.1% SDS, 192 mM glycine, and 25 mM Tris at pH 6.8. Proteins were visualized via staining with Coomassie brilliant blue R-250. the proteins used for molecular weight markers were BenchMark^TM^ prestained Protein Ladder and BenchMark^TM^ XP Western Protein Standard (ThermoFisher Scientific).

After samples were separated on SDS-PAGE gels, they were transferred to Immobilon^TM^-P transfer membranes using standard methods [37]. Membranes were blocked for 1 h at room temperature with 5% nonfat dry milk in Tris-buffered saline with 0.05% Tween 20 (TBS-T). Blots were probed using polyclonal rabbit anti-PBCV-1 (Harlan) antibody overnight with agitation at 4 °C in a blocking buffer. After five washes with TBS-T, membranes were probed with goat anti-rabbit conjugated to horseradish peroxidase (HRP) as the secondary antibody. After an additional five washes with TBS-T, the membranes were developed using an ECL SuperSignal West Pico chemiluminescent substrate (ThermoFisher Scientific). Bands were visualized using a ChemiDoc^TM^ MP Imaging System.

### 2.6. Quantitative Polymerase Chain Reaction (qPCR) Analysis of Virus DNA Abundance

OSy-NE5 and PBCV-1 DNA accumulation kinetics in NC64A and Syngen cells were detected using qPCR by tracking the DNA kinetics of two viral genes PI. Two genes in OSy-NE5 (OS5_104L and OS5_154L) and their homologs in PBCV-1 (A208R and A312L) were selected for this experiment. Primers for these two genes (Appendix A) were designed using Primer-BLAST (http://www.ncbi.nlm.nih.gov/tools/primer-blast, accessed on 18 May 2023), and were aligned against the whole virus and the host genome using BLASTn hits (http://blast.ncbi.nlm.nih.gov, accessed on 18 May 2023).

qPCR experiments were performed in 64-well plates in an Eco Real-Time PCR System (Illumina, San Diego, CA, USA). Reactions were performed in 20 µL volumes comprising 10 µL 2 × GoTaq^®^ qPCR Master Mix, 1 µL of each forward and reverse primer (10 µM), 7 µL of nuclease-free water, and 1 µL of DNA samples. The PCR amplification conditions were 95 °C for 2 min, followed by 40 cycles of 95 °C for 15 s, 60 °C for 60 s, and followed by a dissociation gradient at 95 °C for 15 s, 55 °C for 15 s, and ended with an increase to 95 °C for 15 s to build the melting curve. Melting curves were used to verify the primers pair’s specificity. No-template controls were also included. Gene copies were measured via an absolute quantification using a standard curve for each gene.

## 3. Results

### 3.1. OSy-NE5 Virus Infection Killed Permissive (Syngen 2-3) and Nonpermissive (NC64A) Cells

As reported previously, OSy-NE5 attached to nonpermissive NC64A cells that resulted in apparent initiation of infection but no virus replication [33]. This observation led to the question: what is the fate of NC64A cells after exposure to OSy-NE5? This question was addressed by inoculating Syngen and NC64A cells (1.5 × 10^7^ cells/mL) with OSy-NE5 at low (0.001 PFU/cell) and high (10 PFU/cell) MOIs, and cell viability was evaluated by counting the number of actively growing cells for 15 days at various times PI; OSy-NE5 titers were also monitored using the plaque assay. OSy-NE5 lysed Syngen cell cultures at both low and high MOIs (Figure 1B,C), with the expected increase in virus titers.

However, there was no cell growth in NC64A cells or increase in OSy-NE5 inoculated with a high MOI, suggesting that all the NC64A cells were killed after infection at a high MOI. In contrast, cell growth similar to that of the uninfected controls was observed in NC64A cells infected with OSy-NE5 at a low MOI (Figure 1A–C). However, there was no increase in the OSy-NE5 titer in these NC64A cells. Similar results were obtained with PBCV-1 infection of both cell types, except that PBCV-1 also inhibited the growth of Syngen cells at a low MOI because the cells supported virus replication (results not shown). In summary, OSy-NE5 was unable to replicate in NC64A cells. However, inoculation of NC64A cells with OSy-NE5 resulted in the death of NC64A cells.

### 3.2. OSy-NE5 Virus Infection of Nonpermissive NC64A Cells Prevented Secondary PBCV-1 Infection

A previous study reported that PBCV-1 infection of NC64A cells prevented infection by a second chlorovirus a short time later [38]. Therefore, we challenged NC64A cells with OSy-NE5 and then subsequently with PBCV-1 to determine if there was an exclusion of PBCV-1 infection. Actively growing Syngen and NC64A cells at 1 × 10^7^ cells/mL were inoculated with either OSy-NE5, PBCV-1, or ATCV-1 (as a negative control because it does not attach to or replicate in either NC64A or Syngen, and consequently does not impact their physiologies) viruses at an MOI of 10 PFU/cell for 30 min. Cells were pelleted via centrifugation, and the cells were washed and resuspended two times in MBBM. Cells were then inoculated with PBCV-1 at an MOI of 0.01 PFU/cell and lysates were plaque-assayed on NC64A cells at 96 h PI. As reported in Figure 2, OSy-NE5 infection of Syngen cells prevented subsequent formation of PBCV-1 plaques. Interestingly, similar to the scenario occurring for permissive cells, OSy-NE5 infection of nonpermissive NC64A cells also prevented subsequent PBCV-1 replication. Thus, OSy-NE5 infection prevents PBCV-1 infection of permissive and nonpermissive cells. Control experiments with ATCV-1 or PBCV-1 primary infection followed by PBCV-1 infection did not affect PBCV-1 formation, as expected.

### 3.3. Synthesis of OSy-NE5 Viral Protein in Permissive (Syngen 2-3) and Nonpermissive (NC64A) Cells

The effect of OSy-NE5 infection of Syngen and NC64A cells on viral protein synthesis was examined using SDS-PAGE and Western blotting. The PBCV-1-infected hosts served as positive controls. OSy-NE5 proteins could easily be detected using anti-PBCV-1 antibodies. Samples of infected and uninfected cell lysate proteins were separated and loading equivalents were normalized using SDS-PAGE with Coomassie stain before blotting on a membrane. Immunoblots showed that OSy-NE5 protein levels increased with increasing time following viral infection of NC64A and Syngen cells by 3 h PI (Figure 3A). However, the levels of OSy-NE5 proteins in NC64A cells were lower than their level in Syngen cells after 3 h PI. The increase in the OSy-NE5 protein levels in Syngen cells was similar to that observed in PBCV-1-infected NC64A and Syngen cells (Figure 3B). Therefore, OSy-NE5 transcribed and translated at least some of its genes in both Syngen and NC64A cells.

### 3.4. Ultrastructure Analysis of OSy-NE5 Virus Infection in Permissive and Nonpermissive Cells

To determine why OSy-NE5 does not complete its replication cycle in NC64A cells, OSy-NE5 infections in both the permissive (Syngen) and nonpermissive (NC64A) hosts were examined via transmission electron microscopy. As noted in Figure 4, OSy-NE5 attached to and degraded the cell wall and emptied its contents into both its permissive host (Figure 4A,B) and nonpermissive host (Figure 4C,D), which supports the concept that OSy-NE5 successfully initiates infection of NC64A cells, i.e., both cell types are susceptible to infection. Furthermore, the process appeared to be identical to how PBCV-1 infects NC64A cells, which has previously been described [28,39,40,41]. PBCV-1 attaches to NC64A via a spike structure that is located at one of its pentagonal vertices [41,42]. Immediately after PBCV-1 attachment, the cell wall is degraded at the point of attachment by a virus-associated enzyme [43]. After host cell wall degradation, the PBCV-1 internal membrane fuses with the host membrane [44], facilitating the entry of the viral core materials, DNA, and virion-associated proteins into the cell, leaving an empty capsid on the surface [45]. Virus infection causes rapid depolarization of the host plasma membrane that leads to the inhibition of secondary active transporters, consequently altering cellular solute uptake [46]. OSy-NE5, like PBCV-1, does not encode a recognizable RNA polymerase, so viral DNA and some viral proteins presumably move to be in the proximity of the nucleus like PBCV-1 during infection [28,39,44,47,48,49].

Further ultrastructure analyses showed that the successful infection process of OSy-NE5 in Syngen cells (Figure 5) is very similar to PBCV-1 infection in both Syngen (Appendix A) and NC64A cells [28,39,40,41]. Interestingly, electron microscopic analysis revealed similar events for OSy-NE5 on the nonpermissive host (NC64A), except no mature viral progeny with packaged DNA was observed (Figure 6 and Figure 7).

In its permissive host (Syngen), the first morphological evidence of OSy-NE5 infection was observed at 1 h PI; as the infected host nuclei became deformed and lost their spherical shape as compared to uninfected cells, the nucleolus disintegrated, and the chromatin material condensed (certain areas in the nucleus were heavily stained for DNA leaving empty areas) (Figure 5B). In addition to the changes that occurred in the nucleus, the nucleus and cytoplasmic components were pressed against the chloroplast, leaving a large cytoplasmic area. Similar events were observed in NC64A cells after 1 h of infection with PBCV-1 [39,44]. The preceding modification in the nucleus structure was coincident with extensive degradation of host DNA in PBCV-1-infected NC64A. PBCV-1 encodes for DNA restriction endonucleases and packages them in the mature virion, which helps in degrading the host DNA in the early phase of infection [35]. OSy-NE5 is also predicted to encode about 10 endonucleases [33].

At 3 h PI of Syngen cells, OSy-NE5 capsids assembled in virus assembly centers (VCs) in the cytoplasm, and the cells contained many opaque virus particles (presumably mature virus with DNA) as well as a few empty capsids (Figure 5C). By 6 h PI, the cytoplasm was filled with numerous mature virus particles (Figure 5D) waiting for lysis of the membrane and the wall; however, the empty capsids were restricted to the VC, apparently waiting for DNA packaging, which supports the idea that DNA packaging occurs in the VC. Previous observations suggested that viral proteins were assembled into icosahedral particles before DNA entry [44,50].

In the nonpermissive host (NC64A), there was a delay in the response of the cells to OSy-NE5 infection, and the first morphological evidence of infection was observed at 3 h PI (Figure 6C); the cells deteriorated, and no virus particles were observed up to 12 h PI (Figure 6B–E). However, at 24 h PI, OSy-NE5-empty capsids were observed in about 20% of the infected cells (Figure 6F and Figure 7). It was also interesting that some infected cells with empty capsids burst, releasing some of the imperfect particles (Appendix A).

### 3.5. Quantification of OSy-NE5 Virus DNA Accumulation in Permissive and Nonpermissive Cells

These ultrastructural observations led to the question: were the empty capsids due to a problem in DNA packaging or did OSy-NE5 fail to replicate its DNA in NC64A cells? To address this question, we investigated the accumulation of OSy-NE’s DNA in both nonpermissive (NC64A) and permissive (Syngen) hosts (Figure 8). The PBCV-1-infected hosts served as positive controls. A previous study reported a four-fold increase in virus DNA in PBCV-1-infected NC64A cells at 6 h PI, which resulted from the degradation of host DNA and synthesis of viral DNA (Figure 8D) [45]. Like PBCV-1, infection of OSy-NE5 in its permissive host (Syngen) also increased the DNA in the cell about four-fold compared to the uninfected control (Figure 8A). However, when OSy-NE5 infected the nonpermissive host (NC64A), no increase in the DNA occurred in the infected cells (Figure 8A). This suggested that OSy-NE5 DNA did not replicate in NC64A cells.

To verify this conclusion, two viral DNA genes were quantified based on absolute levels via qPCR and were normalized as the number of gene copies per cell. DNA was isolated from an equal number of cells from OSy-NE5-infected NC64A and Syngen cells at different times following viral infection. DNA from PBCV-1-infected cells was also isolated and used as a positive control. Two genes in OSy-NE5 (OS5_104L and OS5_154L) and their homologs in PBCV-1 (A208R and A312L) were evaluated in the assays as proxies for viral genome abundances. The results indicated that OSy-NE5 gene copies in Syngen-infected cells increased after 1 h PI (Figure 8B,C), which is very similar to what was observed in PBCV-1-infected NC64A and Syngen cells (Figure 8E,F). This indicates that OSy-NE5 and PBCV-1 initiate DNA replication by approximately 1 h PI in their compatible hosts. In contrast, there was no increase in OSy-NE5 gene copies in NC64A-infected cells, which means that OSy-NE5 DNA did not replicate in NC64A cells. These experiments were extended to 24 h PI, and still there was no increase in OSy-NE5 DNA in NC64A cells [51]. Therefore, the results of these experiments support that restriction of OSy-NE5 in NC64A cells has to do with their inability to replicate viral DNA. This suggests that one or more of the enzymes involved in virus DNA replication are inhibited.

## 4. Conclusions

Results from this study and a previous paper [33] establish that chlorovirus OSy-NE5 can attach to and initiate infection, including releasing the virus genome into the cytoplasm of the nonpermissive host (*Chlorella variabilis* NC64A). This infection killed the cells and resulted in the synthesis of at least some early virus gene products, leading to some empty virus capsids by 24 h PI. However, the infected cells were unable to produce infectious OSy-NE5 viruses because the cells were unable to replicate the viral genome. All of our previous experiments aiming to detect resistance to the chloroviruses demonstrated changes in the host receptor to the virus, which is located in the host cell wall. Taken together, these data show an emerging paradigm in the biology of chlorovirus–host interaction, suggesting that chlorovirus infection resistance is multidimensional as it might originate either initially by blocking the virus attachment by altering the receptor sites at an early stage of infection, or intracellularly by inhibiting virus DNA replication, which leads to an inefficient infection.

## Figures and Tables

**Figure 1 viruses-15-01341-f001:**
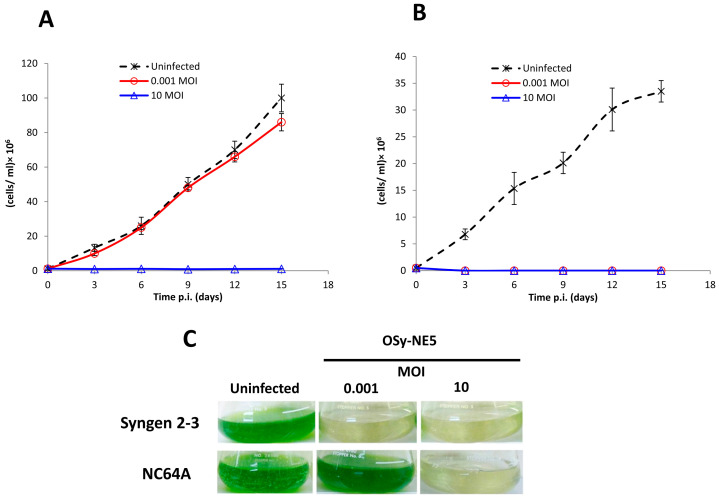
Growth of NC64A cells (**A**) and Syngen cells (**B**) after infection with OSy-NE5 at low MOI (0.001 PFU/cell) and high MOI (10 PFU/cell). Uninfected cells served as a negative control. The growth was monitored for 15 days PI and is expressed as number of cells/mL. (**C**) Images of the untreated and OSy-NE5-treated algal cultures.

**Figure 2 viruses-15-01341-f002:**
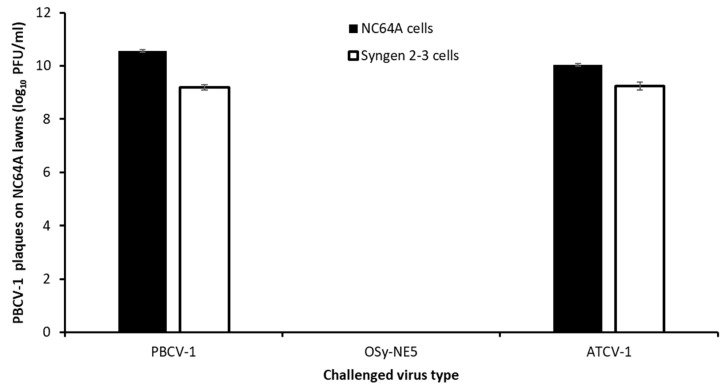
Initial OSy-NE5 infection inhibits subsequent PBCV-1 replication on NC64A and Syngen cells. The number of PBCV-1 plaques on NC64A lawns after 96 h (y-axis) following initial exposures to viruses (MOI = 10 PFU/cell) PBCV-1, OSy-NE5, or ATCV-1 on NC64A and Syngen cells followed by a second exposure to PBCV-1 (x-axis).

**Figure 3 viruses-15-01341-f003:**
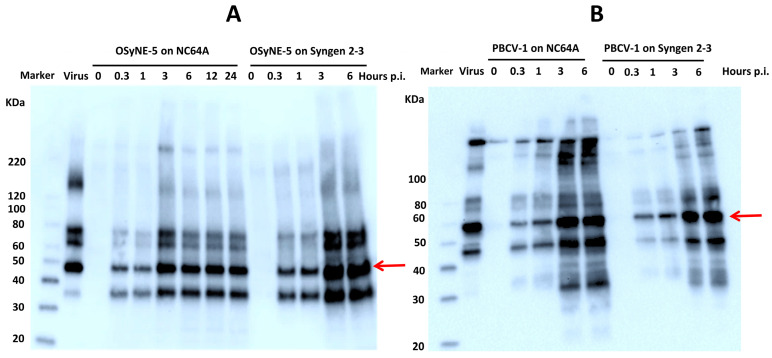
Accumulation of virus protein in both permissive and nonpermissive hosts. (**A**) Western blot analysis showing dynamics of OSy-NE5 protein accumulation isolated from uninfected (0) and OSy-NE5-infected NC64A cells at 20 min, 1, 3-, 6-, 12-, and 24 h PI and Syngen cells at 0, 20 min, 1, 3, and 6 hr PI. (**B**) Western blot analysis showing dynamics of PBCV-1 protein accumulation isolated from uninfected (0) and PBCV-1-infected NC64A and Syngen cells at 20 min, 1, 3, 6 h PI. The red arrows indicate the viruses’ major capsid proteins. Blots were probed using anti-PBCV-1 protein antibody. Equal loading of protein occurred for each sample.

**Figure 4 viruses-15-01341-f004:**
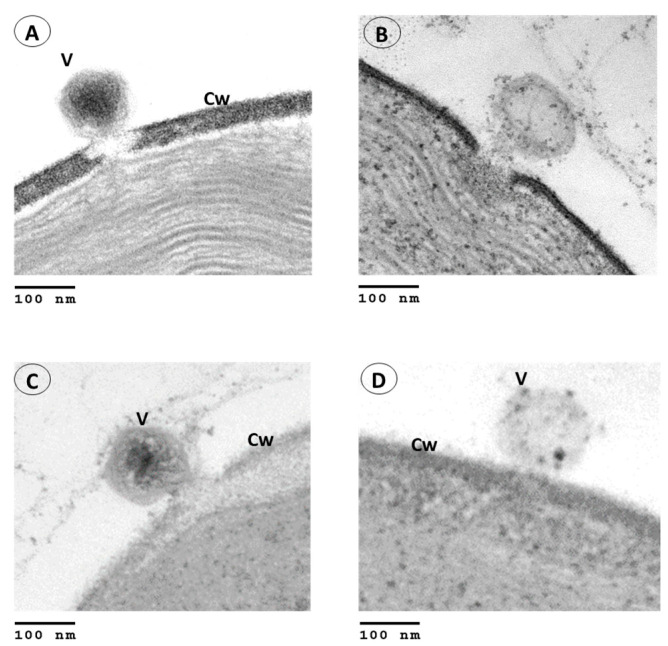
Transmission electron micrographs of the infection process of *C. variabilis* Syngen 2-3 and *C. variabilis* NC64A cells by OSy-NE5. (**A**,**B**) Attachment and release of OSy-NE5 DNA into Syngen cells. (**C**,**D**) Attachment and release of OSy-NE5 DNA into NC64A cells. Symbols: cell wall, Cw, and virus particle, V. Scale bars, 100 nm.

**Figure 5 viruses-15-01341-f005:**
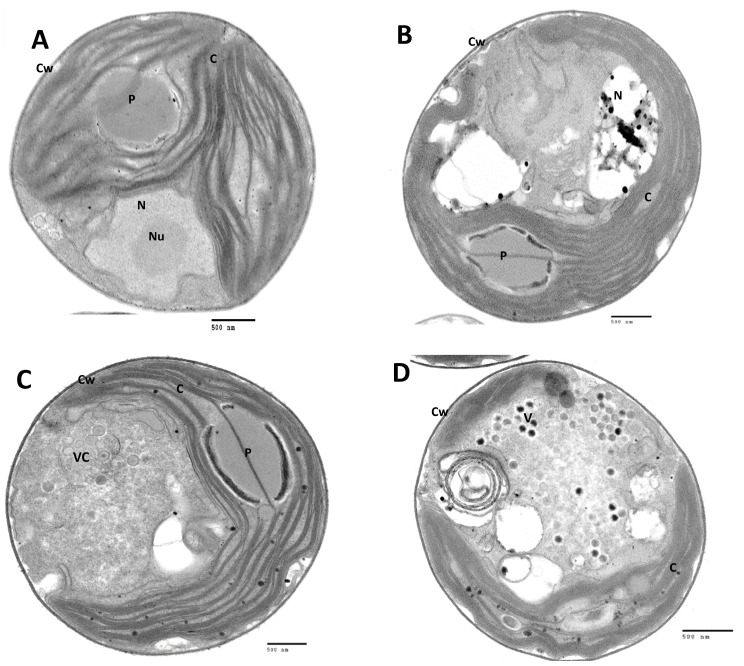
Transmission electron micrographs of OSy-NE5-infected and uninfected *C. variabilis* Syngen 2-3. (**A**) uninfected Syngen 2-3 cell, (**B**–**D**) 1 h, 3 h, and 6, respectively, after infection with OSy-NE5. Mature virus particles were formed at 6 h PI in Syngen cells. Symbols: chloroplast, C, pyrenoid, P, nucleus, N, nucleolus, Nu, cell wall, Cw, virus assembly center, VC, and virus particles, V. Scale bars, 500 nm.

**Figure 6 viruses-15-01341-f006:**
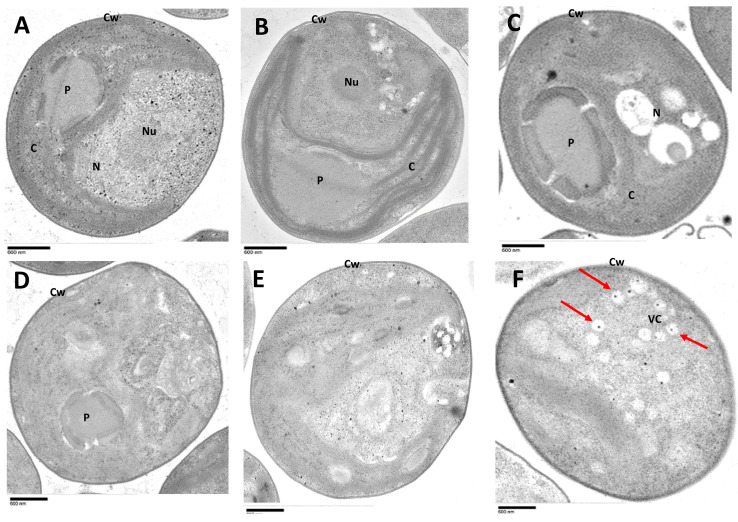
Transmission electron micrographs of OSyNE-5-infected and uninfected *C. variabilis* NC64A. (**A**) Uninfected NC64A cell (**B**–**F**) 1, 3, 6, 12, and 24 h, respectively, after infection with OSy-NE5. Empty OSy-NE5 capsids were formed after 24 h PI, as shown by red arrows. Symbols: chloroplast, C, pyrenoid, P, nucleus, N, nucleolus, Nu, cell wall, Cw, virus assembly center, VC, and virus particles, V. Scale bars, 600 nm.

**Figure 7 viruses-15-01341-f007:**
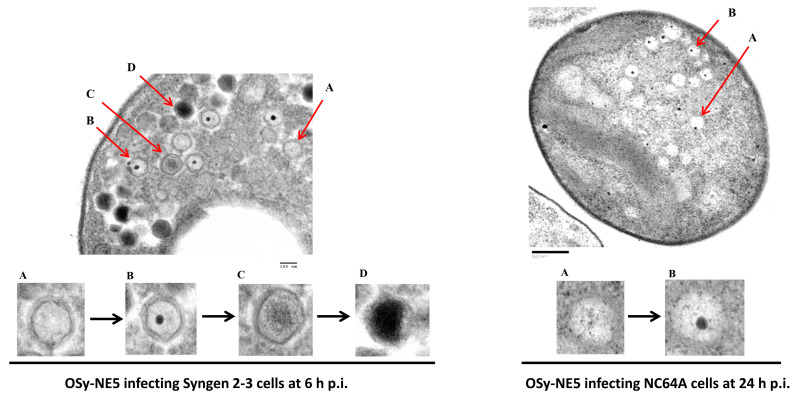
Close-up view of OSy-NE5 infection process of Syngen cells and NC64A cells. The left panel shows a Syngen 2-3 cell at 6 h PI with OSy-NE5, while the right panel shows a NC64A cell at 24 h PI with OSy-NE5. The red arrows are virus particles at different stages of development. Magnified sections of these stages are shown in the bottom panels (A–D): (A) viral empty capsid; (B) presumed initiation of DNA packaging; (C) partially packaged (mature) virus; (D) fully packaged virus particle. Note that the capsids in the NC64A-infected cells are empty of viral DNA. Scale bars, 100 nm (**left** panel) and 500 nm (**right** panel).

**Figure 8 viruses-15-01341-f008:**
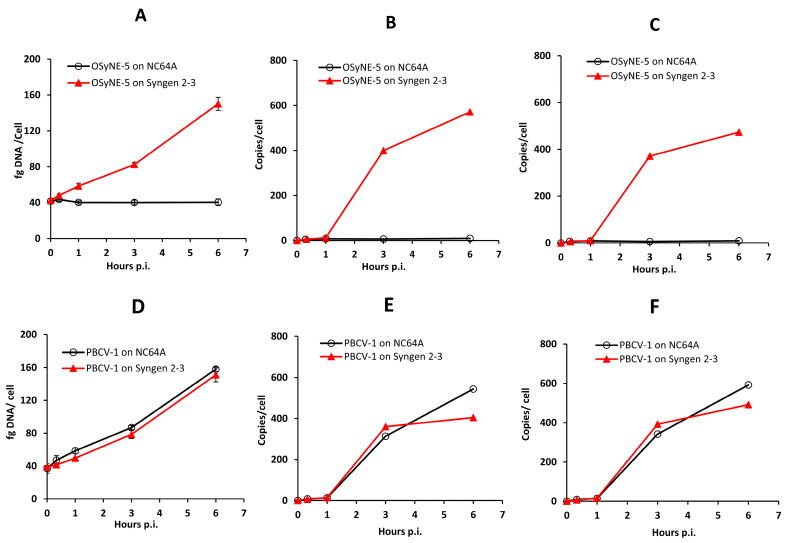
DNA dynamics of OSy-NE5 in Syngen and NC64A cells. Total DNA concentrations in OSy-NE5-infected NC64A (**A**) and Syngen (**D**) cells. Specific gene replication of OSy-NE5 on NC64A and Syngen cells was monitored using qPCR analysis by measuring the kinetics of two genes at various times PI. Two OSy-NE5 genes, OS5_104L (**B**) and OS5_154L (**C**), and their corresponding homologs in PBCV-1 A208R (**E**) and A312L (**F**), respectively, were selected for this experiment. The results shown are the quantification of each gene copies/cell established using a standard curve. PBCV-1 served as a positive control that shows gene replication in both hosts.

## Data Availability

All processed data used in the manuscript are presented in the manuscript figures, tables and the Appendix A.

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
