# Peer review of "Viral DNA Accumulation Regulates Replication Efficiency of Chlorovirus OSy-NE5 in Two Closely Related Chlorella variabilis Strains"

_viruses, 2023, doi:10.3390/v15061341_

Round 1

Reviewer 1 Report

Functional studies on giant viruses are very rare,
and it is a pleasure to receive such a manuscript for review,
which constitutes an important advance in the field.
I have some minor comments and suggestions for improvments

Line 35: I am not sure that "however" is the right way to
start a sentence that is about interspecificity following one about
intraspecificity. Perhaps "on the other hand"

Line 43-45 and Ref 8:
It would have been nice to have individual references for each
of the cited mechanisms of resistance. Reference 8 is quite ancient,
and should be complemented.

Line 51:
reference 9 is about herpesvirus and not PpV!

Line 55: Starting from Ref 10, the numbering is false (should ne n+1)

Section 3.2, line 227-240:
It would have been nice to have/know the results for the lower
MOI (0.001 PFU/cell) as shown in Figure 1A, as it looks that
at MOI= 10 PFU/cell the host cells are killed, hence have no
way to respond to a subsequent infection by another virus.

Line 357-358: May be the similarity with the nucleus deformation
seen in the case of Marseillevirus (Noumeavirus) should be
mentionned (this virus encodes, but does not embark its RNA polymerase
iin its particle).

Line 364-368: Optional: It would be nice to have some information on
host's DNA degradation for OSy-NE5 infections in
Syngen and/or NC64A cells.

Line 378-379: Useful to precise "instead of 1 h"

Line 456: use ":" instead of ";" ?

4. Conclusion.
Line 503-507: as it is an important emerging paradigm, it is a
little bit frustration not to see a few speculations about which
mechanism and/or gene could be involved. Any possible clues from
genomic differences between OSy-NE5 and PBCV-1 ?
Any from transcriptomics? At this point speculations are welcome, as
well as possible future developments.

Line 526: Jean-Michel Claverie (instead of "Michele"

Author Response

Dear Dr. Zhai,

We want to thank the reviewers for their nice comments on the manuscript entitled “Viral DNA Accumulation Regulates Replication Efficiency of Chlorovirus OSy-NE5 in Two Closely Related Chlorella variabilis strains” that was reviewed for Viruses. We have responded to the comments of reviewer 1 (below) and reviewer 2 did not have any comments. We hope the manuscript is now suitable for publication in Viruses.

Sincerely,

James L. Van Etten

William B. Allington Distinguished

Professor in Plant Pathology

University of Nebraska – Lincoln

Lincoln, NE 68683-0900

Email: jvanetten1@unl.edu

Reviewer#1

Comments and Suggestions for Authors

Functional studies on giant viruses are very rare, and it is a pleasure to receive such a manuscript for review, which constitutes an important advance in the field. I have some minor comments and suggestions for improvements.

Reply: Thanks for the nice comments.

Line 35: I am not sure that "however" is the right way to
start a sentence that is about interspecificity following one about
intraspecificity. Perhaps "on the other hand"

Reply: The word has been changed.

Line 43-45 and Ref 8:
It would have been nice to have individual references for each
of the cited mechanisms of resistance. Reference 8 is quite ancient,
and should be complemented.

Reply: we have added a recent reference, please check reference# 8 in the edited MS.

Line 51:
reference 9 is about herpesvirus and not PpV!

Reply: It was because of the mis ordering of the references, it has been corrected.

Line 55: Starting from Ref 10, the numbering is false (should ne n+1)

Reply: Thank you for noticing , we have re numbered the references.

Section 3.2, line 227-240:It would have been nice to have/know the results for the lower
MOI (0.001 PFU/cell) as shown in Figure 1A, as it looks that
at MOI= 10 PFU/cell the host cells are killed, hence have no
way to respond to a subsequent infection by another virus.

Reply: The results of the lower MOI (0.001 PFU/cell) are shown in Fig. 1A – the red line. Like the higher MOI, the Osy-NE5 virus kills the NC64A cells. However, because there is so few viruses, only a few NC64A cells are killed and so the population of NC64A grows, almost like if there is no virus added.

Line 357-358: May be the similarity with the nucleus deformation
seen in the case of Marseillevirus (Noumeavirus) should be
mentionned (this virus encodes, but does not embark its RNA polymerase
iin its particle).

Reply: This is an interesting comment and we will keep it in mine for future experiments. However, we prefer not to say anything at the present time because I looked up several publications including two recent reviews on Marseilleviruses and neither one of the recent reviews said anything about changes in the morphology of the nucleus during virus infection.

Line 364-368: Optional: It would be nice to have some information on
host's DNA degradation for OSy-NE5 infections in
Syngen and/or NC64A cells.

Reply: This is a good suggestion and may be looked at in the future using pulsed field gel electrophoresis as well as trying to determine if virus Osy-NE5 actually encodes and packages a restriction endonuclease(s).

Line 378-379: Useful to precise "instead of 1 h"

Reply: we are not sure what this means.

Line 456: use ":" instead of ";" ?

Reply: It has been changed.

4. Conclusion.
Line 503-507: as it is an important emerging paradigm, it is a
little bit frustration not to see a few speculations about which
mechanism and/or gene could be involved. Any possible clues from
genomic differences between OSy-NE5 and PBCV-1 ?
Any from transcriptomics? At this point speculations are welcome, as
well as possible future developments.

Reply: We have added the following sentence at the end of the Results/Discussion section. “This suggests that one or more of the enzymes involved in virus DNA replication are inhibited.”

Line 526: Jean-Michel Claverie (instead of "Michele")

Reply: It has been changed.

Reviewer 2 Report

In my opinion, the paper is a very good one. It combines detailed logical planning of experiments, excellent methodical execution, and excellent results. I especially want to note the use of transmission electron microscopy, which allowed reliable visualization of the obtained results and conclusions. Recently, the majority of works in the field of virology have used predominantly molecular genetic methods. In this work, an integrated approach was used.

Author Response

Dear Dr. Zhai,

We want to thank the reviewers for their nice comments on the manuscript entitled “Viral DNA Accumulation Regulates Replication Efficiency of Chlorovirus OSy-NE5 in Two Closely Related Chlorella variabilis strains” that was reviewed for Viruses. We have responded to the comments of reviewer 1 (below) and reviewer 2 did not have any comments. We hope the manuscript is now suitable for publication in Viruses.

Sincerely,

James L. Van Etten

William B. Allington Distinguished

Professor in Plant Pathology

University of Nebraska – Lincoln

Lincoln, NE 68683-0900

Email: jvanetten1@unl.edu

Reviewer#2

Comments and Suggestions for Authors

In my opinion, the paper is a very good one. It combines detailed logical planning of experiments, excellent methodical execution, and excellent results. I especially want to note the use of transmission electron microscopy, which allowed reliable visualization of the obtained results and conclusions. Recently, the majority of works in the field of virology have used predominantly molecular genetic methods. In this work, an integrated approach was used.

Reply: Thanks for the nice comments
